# Peptide Assembly of Al/CuO Nanothermite for Enhanced Reactivity of Nanoaluminum Particles

**DOI:** 10.3390/ijms23148054

**Published:** 2022-07-21

**Authors:** Miaomiao Jin, Zhanxin Song, Wei Liu, Zilu Zhou, Guozhen Wang, Mo Xian

**Affiliations:** 1Key Laboratory of Biobased Materials, Qingdao Institute of Bioenergy and Bioprocess Technology, Chinese Academy of Sciences, Qingdao 266101, China; songzx@qibebt.ac.cn (Z.S.); zhouzl@qibebt.ac.cn (Z.Z.); wanggz@qibebt.ac.cn (G.W.); 2Institute of Corrosion Science and Technology, Guangzhou 510530, China; liuwei@qibebt.ac.cn

**Keywords:** biological self-assembly, thermal analysis, peptide, Al nanoparticles stability

## Abstract

Biological self-assembly procedures, which are generally carried out in an aqueous solution, have been found to be the most promising method for directing the fabrication of diverse nanothermites, including Al/CuO nanothermite. However, the aqueous environment in which Al nanoparticles self-assemble has an impact on their stability. We show that using a peptide to self-assemble Al or CuO nanoparticles considerably improves their durability in phosphate buffer aqueous solution, with Al and CuO nanoparticles remaining intact in aqueous solution for over 2 weeks with minimal changes in the structure. When peptide-assembled Al/CuO nanothermite was compared with a physically mixed sample in phosphate buffer for 30 min, the energy release of the former was higher by 26%. Furthermore, the energy release of peptide-assembled Al/CuO nanocomposite in phosphate buffer showed a 6% reduction by Day 7, while that of the peptide-assembled Al/CuO nanocomposite in ultrapure water was reduced by 75%. Taken together, our study provides an easy method for keeping the thermal activity of Al/CuO nanothermite assembled in aqueous solution.

## 1. Introduction

Nanothermites, comprising oxidizer and metal fuel existing at the nanoscale, are quite important among energetic materials, on account of their high energy density and rapid energy release [1,2,3,4,5]. As a hot research topic in recent years, nanothermites have found uses in a multitude of applications. These include their uses as detonation initiators [6], pyrotechnics [7], aerospace devices [8,9], airbag ignition materials [10], and in biomedical applications [11,12].

A thermal reaction is a typical solid-state reaction having highly exothermic and self-propagating characteristics [13]. The interfacial diffusion generally determines the heat and mass transfer, which is a fundamental process affecting combustion efficiency and energy release [14]. In nanothermites, oxidizers and fuel contact each other at the nanoscale, the intimacy between oxidizers and fuel reduces, which in turn causes an extraordinary burning rate and an outstanding efficiency of energy release. Due to the ease with which energetic nanothermites aggregate, it is difficult to considerably improve or accurately manage their performance. In this context, a variety of attempts such as the sol–gel method [15], electrophoretic deposition [16], electrostatic assembly [17], and magnetron sputtering [14] have been made to explore the strategies to prepare nanothermites. Although the methods described above can be implemented simply in a variety of domains, they are not suited for large-scale applications. Biological self-assembly approaches have been regarded as the most promising technique for controlling nanothermite self-assembly [18,19,20,21]. By using the unique tight contact between the nanoparticles and the peptide, our group has developed a novel biological self-assembly approach for manufacturing Al and CuO nanocomposites [22]. The approach has additional benefits including: improved mixing uniformity, mild reaction conditions, convenient operation, and eco-friendliness.

Aluminum nanoparticles (Al NPs) are commonly utilized as fuel in nanothermite due to its inexpensive cost, availability, energy density, and extraction efficiency [23,24,25]. Al NPs have substantially higher chemical reactivity than other metal nanothermites, generating a 2–5 nm oxide layer when exposed to air [26,27,28]. In the air and the majority of organic solvents, the oxide layer may prevent additional corrosion, but it does not protect them from additional oxidation in an aqueous solution [29]. Because many applications of nanothermites (nano-Al NPs-based energetic materials) involve a watery or humid environment, they pose a significant problem [18,30]. Although peptides have been successfully employed to self-assemble Al and CuO nanothermites with several benefits, further research is needed to appropriately stop the oxidation of Al NPs in an aqueous environment.

In this regard, this paper describes a peptide-based approach for assembling Al NPs and CuO NPs in phosphate buffer, with Al NPs remaining impressively stable for over two weeks with only a few structural alterations. There is only limited oxide development in comparison with that seen in bare Al NPs. To achieve these objectives, we used dynamic light scattering (DLS), zeta potential measurements, transmission electron microscopy (TEM), pH, and heat release to investigate the self-assembly of Al/CuO nanothermite.

## 2. Results and Discussion

### 2.1. Aqueous Stability of Al NPs

Al NPs tend to lack stability in an aqueous environment; therefore, the aqueous stability of Al NPs should be investigated in order to suggest effective ways to prevent them from further oxidation. In an attempt to investigate the stability of Al NPs in an aqueous solution, we conducted a TEM analysis. Al NPs were dispersed in phosphate buffer (PBS, 0.01 M, pH 7.0) or ultrapure water and subjected to sonication for 11 min. Subsequently, Al NPs were allowed to stir gently at room temperature. Stirring was continued to allow the suspension of nanoparticles in solution, thus preventing their sedimentation.

Al NPs were much more stable under PBS conditions than in ultrapure water, as demonstrated in Figure 1. After around 3 days, the Al NPs in ultrapure water began to degrade, and filamentary oxide structures began developing on the surfaces. The Al NPs were clearly oxidized after 14 days. No metallic Al NPs were observed in the TEM images after 30 days (Figure 1A). The Al NPs, on the other hand, had smooth surfaces with no visible oxide development after the same duration of PBS treatment (Figure 1B). These results demonstrate that the PBS environment can provide better protection for Al NPs against oxidation during assembly in an aqueous solution.

XRD analysis was used to evaluate the Al NPs dispersed in the aqueous solution used in this investigation, as shown in Figure 2. The signals of Al vanished after being dispersed in ultrapure water, as shown in Figure 2A, while substantial signals of Al(OH)_3_ appeared [28] based on the main peaks at 18.825°, 20.399°, 27.857°, 40.566°, 53.11°, 57.557°, 63.832°, and 70.6° indexed as (001), (020), (111), (201), (202), (222), (330), and (−133), suggesting that the Al NPs were oxidized. Figure 2B at 38.4°, 44.7°, 65.1°, and 78.2° correspond to the (111), (200), (220), and (311) crystal planes of Al (JCPDS 04-0787). The XRD spectra showed very strong diffractions from the Al NPs, suggesting that the PBS conditions can provide better protection for Al NPs against oxidation.

### 2.2. Aqueous Stability of Al/CuO NPs

Al/CuO nanocomposites were assembled in an aqueous solution using peptides. In order to test the efficacy of peptide-assembled Al/CuO nanocomposites in PBS solution and the oxidation resistance of Al NPs, four nanocomposites were produced under various settings. Additionally, the self-assembly conditions were as follows: phosphate buffer (PBS, 0.01 M, pH 7.0) with peptide SH-25 (30 μM), ultrapure water with the peptide SH-25 (30 μM), phosphate buffer (PBS, 0.01 M, pH 7.0) without peptide SH-25, and ultrapure water without the peptide SH-25.

The micromorphology of Al and CuO combined in various solutions is shown in Figure 3. After about 2 days, the Al NPs in the H_2_O/Al/CuO comixture began to degrade. On Day 3, nodules began to build oxide structures on their surfaces. By Day 7, Al NPs were visibly oxidized, and spherical Al NPs had vanished, leaving the oxidized areas flocculent. However, after about 7 days, Al NPs in the H_2_O/SH-25/Al/CuO comixture began to degrade. We previously showed that SH-25 had a particular binding impact on CuO NPs and Al NPs in a prior investigation [22]. As a result of the minor protection of SH-25 in ultrapure water, Al NPs in H_2_O/SH-25/Al/CuO comixture were slightly oxidized. Al NPs in PBS/SH-25/Al/CuO and PBS/Al/CuO comixtures, on the other hand, showed no noticeable alterations in micromorphology. It appears that adding SH-25 to the PBS solution did not affect the aqueous stability of Al NPs. Furthermore, TEM images revealed that SH-25-directed Al/CuO nanocomposites were covered with each other and had rather uniform distributions. This implies that using a peptide assembly method can improve the advantage of phosphate buffer (PBS, 0.01 M, pH 7.0) in reducing Al oxidation and increasing the degree of homogenous intermixing between the Al and CuO components.

To further study the protective interactions between them, the pH of these systems was tested during the time of this study (Figure 4). A variation of pH was observed for the aqueous solution in the H_2_O/SH-25/Al/CuO and H_2_O/Al/CuO comixture system due to the slow oxidation of Al NPs. The pH increased from 5.1 to 7.0 within 7 days when the solution was in H_2_O. In the H_2_O/SH-25/Al/CuO system, the pH increased from the original 5.0 to 5.7. A higher pH change was observed in the H_2_O/Al/CuO system, indicating that the oxidization of Al NPs took place at a faster rate, which coincides with the results obtained from TEM.

DLS and zeta-potential measurements were used to track the features of CuO and Al NPs as they were assembled [21,31].

DLS was used to explore the link between particle size and time in order to demonstrate the ability of SH-25 to guide the assembly of Al and CuO NPs in PBS. The hydrodynamic diameters of various CuO and Al specimens combined in identical stoichiometric ratios are shown in Figure 5. In the first few minutes, there was a rapid size shift in PBS/SH-25/Al/CuO. Within 1 h, the particle size expanded to a micron-scale size from an average size of 500–600 nm, and the size continuously increased over time. Other samples, on the other hand, had no discernible alterations in their hydrodynamic diameters. These findings show that SH-25 in PBS may be used to drive the assembly of Al and CuO via particular molecular recognition.

The zeta-potential values of the corresponding systems were investigated to learn more about their ζ interactions (Figure 6). In 2 days, the absolute value of the zeta potentials of the PBS/Al/CuO comixture, H_2_O/Al/CuO comixture, and H_2_O/SH-25/Al/CuO was above 30 mV (the absolute value of the zeta potentials slightly decreased owing to the oxidation of Al NPs), suggesting the relative stability of these systems, with the mean particle dimensions manifesting no increment. The addition of peptide SH-25 to CuO and Al in a PBS dispersion, however, resulted in a significant reduction in the absolute zeta potential of the CuO and Al-based nanocomposite, which decreased to about 10 mV, demonstrating the effective conjugation of CuO and Al NPs with SH-25.

All of these findings support the idea that SH-25 in PBS operated as an adhesive layering between the particles of CuO and Al, causing fast agglomeration as well as a noticeable shift in the zeta potential and size of PBS/SH-25/Al/CuO. SH-25 dissolved in water, however, failed to cause the assembly of Al and CuO particles. One possible reason might be the change in the secondary structure of SH-25 in H_2_O (pH < 7.0), which subsequently affected the binding affinity [32].

### 2.3. Thermal Analysis

To obtain a better understanding of the effect of the peptide-assembled materials, the thermite reaction characteristics of samples were investigated by DSC (Figure 7). Table 1 shows the energy release observed for the H_2_O/Al/CuO comixture-y, H_2_O/SH-25/Al/CuO-y, PBS/Al/CuO comixture-y, and PBS/SH-25/Al/CuO-y nanothermites at a heating rate of 10 °C/min, where the letter y represents the assembled time. As shown in Table 1, the PBS/SH-25/Al/CuO-30 min had the largest heat output owing to the peptide-based assembly, which was precisely equal to 1785 J/g. The duration of the induced assembly process was extended to 7 d to examine the effect of the PBS of Al/CuO on their reactivity. In ultrapure water, Al NPs gradually transformed to Al(OH)_3_ phase with the total heat energy gradually decreasing. Compared with the H_2_O/SH-25/Al/CuO-30 min, the total heat energy of the H_2_O/SH-25/Al/CuO-7d retained was about 25%. In contrast, PBS/SH-25/Al/CuO did not manifest any significant change in their total heat energy. Indeed, the total heat energy of the PBS/SH-25/Al/CuO-30 min and PBS/SH-25/Al/CuO-7d were 1785 and 1675 J/g, respectively, and the energy retained was as high as 94%. These results show that the nanothermites assembled by the peptide in phosphate buffer aqueous would improve the stability of Al/CuO nanocomposite.

## 3. Materials and Methods

### 3.1. Materials

Nano Material Engineering Company (Jiaozuo, China) provided aluminum NPs with an average diameter of 100 nm and a thin alumina coating on the surface of 4.3± 0.3 nm. Shanghai yunfu Nanotechnology Co., Ltd. (Shanghai, China) provided copper oxide NPs with an average diameter of 20 nm. Beijing Solarbio Science & Technology Co., Ltd. (Beijing, China) provided phosphate buffer (0.01 M, pH 7.0) (PBS). GL Biochem (Shanghai, China) Ltd. provided peptides SH-25 (Ac-STEARATTLTACDAYGGGGHHHHHH-NH_2_) (purity > 95%).

### 3.2. Methods

The target peptide sequences of Al and CuO-specific peptides were determined using the techniques specified in the reference [22], respectively. The sequences of STEARATTLTACDAY and HHHHHH served as linkers to anchor Al and CuO, respectively, and the GGGG sequence was the spacer.

Preparation of the peptide-modified Al/CuO nanocomposites: Al NPs (5.8 mg) and CuO NPs (16.2 mg) was suspended in a 30 mL peptide solution (30 μM was solubilized in 0.01 M phosphate buffer, pH 7.0) and sonicated for 11 min at 100 W, with 2 s pulses separated by 1 s, utilizing a SCIENTZ-IID ultrasonic probe system. For a period of time, the solution was allowed to incubate on the rotator at room temperature. The combined system was subsequently subjected to centrifugation at 10,000 rpm for 10 min and rinsed thrice with ultrapure water. The peptide-assembled energetic Al/CuO nanocomposite PBS/SH-25/Al/CuO was obtained following drying at 30 °C for two days.

Suspending Al NPs and CuO NPs in 30 mL peptide solution (30 M, dissolved in ultrapure water) and sonicating for 11 min yielded peptide-modified Al/CuO nanocomposites in ultrapure water (referred to as H_2_O/SH-25/Al/CuO hereafter). The following steps were the same as before.

The physically mixed sample 1 (hereafter referred to as PBS/Al/CuO comixture) was made by sonicating CuO NPs and Al NPs in a phosphate buffer (PBS, 0.01 M, pH 7.0) in the absence of peptide for 11 min. The subsequent steps were similar to before.

The physically mixed sample 2 (hereafter referred to as H_2_O/Al/CuO comixture) was made by sonicating CuO NPs and Al NPs in ultrapure water without the peptide for 11 min. The following steps were the same as before.

### 3.3. Characterization

A Malvern Zetasizer NanoZS90 instrument was used to investigate the assembly behavior of Al/CuO nanocomposites and the corresponding zeta potentials were determined. Following the incubation period, DLS and ζ measurements were conducted. Thermo gravimetric analysis (TGA) and differential scanning calorimetry (DSC) was used to assess the materials’ thermal performance in a nitrogen environment (50 mL/min) from ambient temperature to 900 °C at a 10 °C/min heating rate using the Mettler Toledo Simultaneous Thermal Analysis TGA/DSC 3+. The samples’ morphological characteristics and elemental compositions were investigated using a TEM (JEM-2100) in combination with EDS and SEM (Hitachi S4800). X-ray diffraction (XRD, D8-Advance X-ray difframeter, Bruker, Germany) was carried out to investigate the crystal phase of the samples after they were disseminated in solution.

## 4. Conclusions

In conclusion, a peptide in an aqueous phosphate buffer was used to direct the self-assembly of Al or CuO-based nanothermites. During 2 weeks in an aqueous solution, this technique was found to greatly enhance the stability of Al NPs in an aqueous solution, avoiding oxidation and conserving their characteristics, with the preserved energy reaching a value of 94% by Day 7. The findings of this study avoid oxidation of aluminum during assembly, which offers new insights for assembling nanothermites using a biological scaffold and a water-based system.

## Figures and Tables

**Figure 1 ijms-23-08054-f001:**
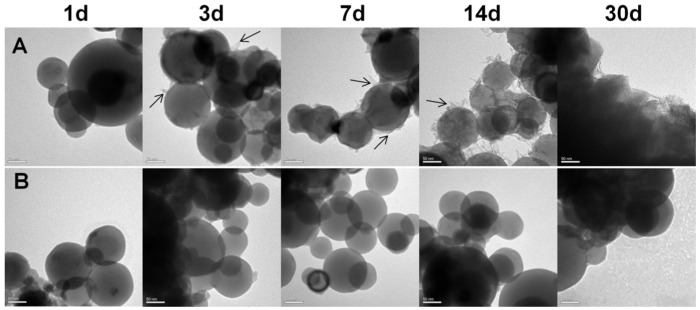
Investigation of oxidation of Al NPs dispersed in aqueous solution at room temperature. (**A**) TEM micrographs of Al NPs dispersed in ultrapure water. (**B**) TEM micrographs of Al NPs dispersed in phosphate buffer (PBS, 0.01 M, pH 7.0). Scale bars are 50 nm.

**Figure 2 ijms-23-08054-f002:**
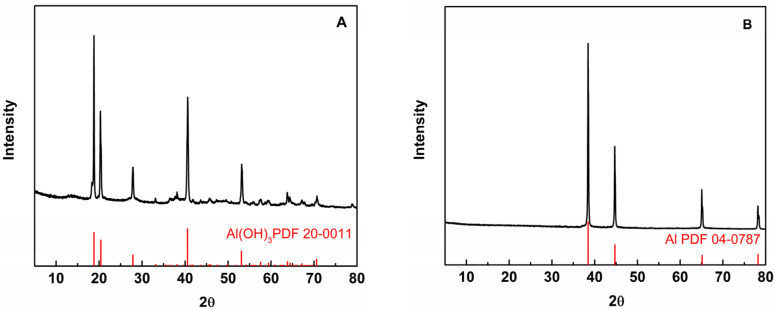
Investigation of oxidation of Al NPs dispersed in aqueous solution at room temperature. (**A**) XRD of Al NPs dispersed in ultrapure water. (**B**) XRD of Al NPs dispersed in phosphate buffer (PBS, 0.01 M, pH 7.0).

**Figure 3 ijms-23-08054-f003:**
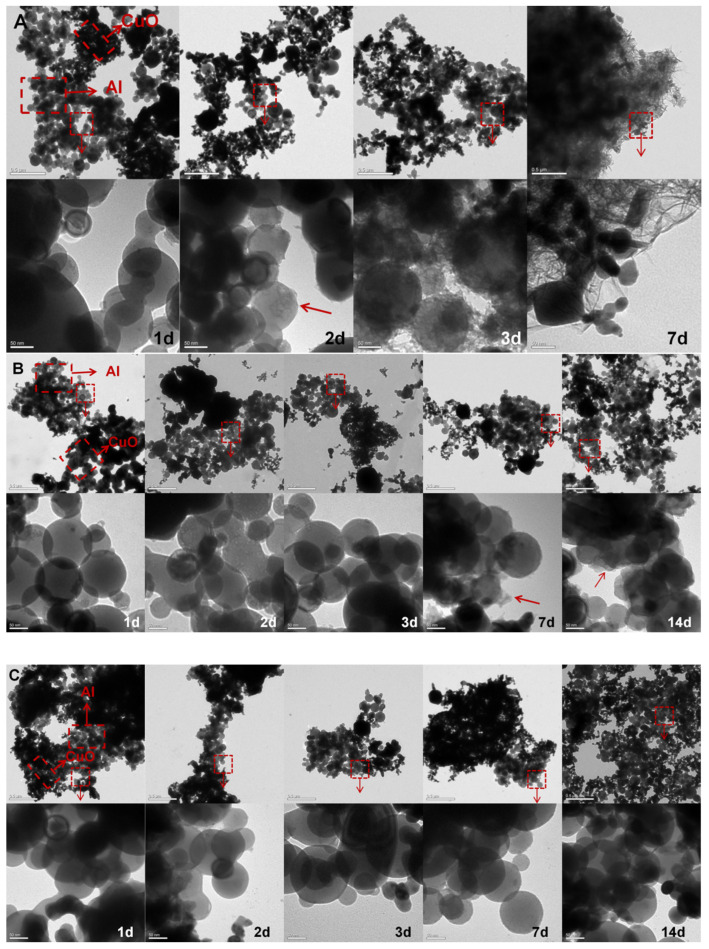
Room-temperature oxidation study of Al and CuO dispersed in an aqueous solution. (**A**) TEM micrographs of H_2_O/Al/CuO comixture at regular intervals during H_2_O exposure. (**B**) TEM micrographs of H_2_O/SH-25/Al/CuO at regular intervals during H_2_O with the peptide SH-25 exposure. (**C**) TEM micrographs of PBS/Al/CuO comixture at regular intervals during PBS exposure. (**D**) TEM micrographs of PBS/SH-25/Al/CuO at regular intervals during PBS with the peptide SH-25 exposure.

**Figure 4 ijms-23-08054-f004:**
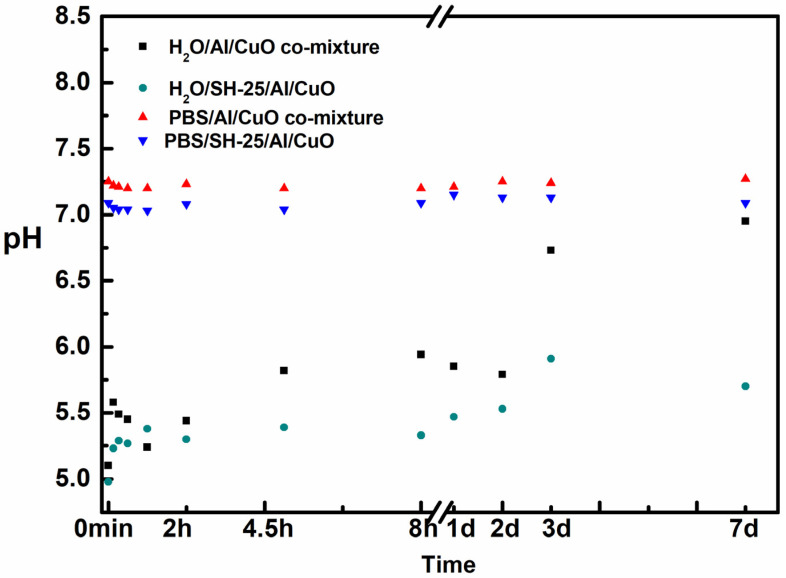
Investigation of the pH solution at room temperature.

**Figure 5 ijms-23-08054-f005:**
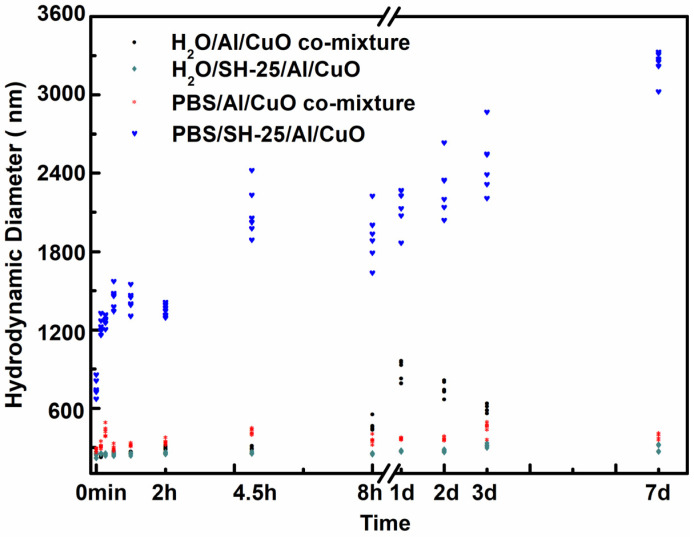
Impact of assembly time on the hydrodynamic diameter. The red dots depict the PBS/Al/CuO comixture. The blue dots denote the PBS/SH-25/Al/CuO. The black dots represent the H_2_O/Al/CuO comixture (owing to the oxidation of Al NPs and the substantial change in particle size, the hydrodynamic diameter was not given on the 7th day). The cyan dots represent H_2_O/SH-25/Al/CuO.

**Figure 6 ijms-23-08054-f006:**
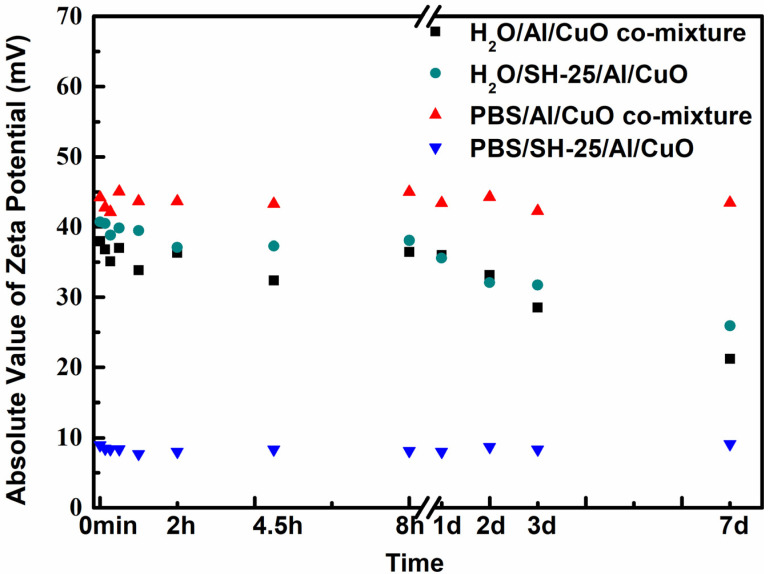
Impact of assembly time on the zeta potentials. PBS/Al/CuO comixture is represented by the red dots. PBS/SH-25/Al/CuO is represented by the blue dots. The black dots represent the H_2_O/Al/CuO comixture. The cyan dots represent H_2_O/SH-25/Al/CuO.

**Figure 7 ijms-23-08054-f007:**
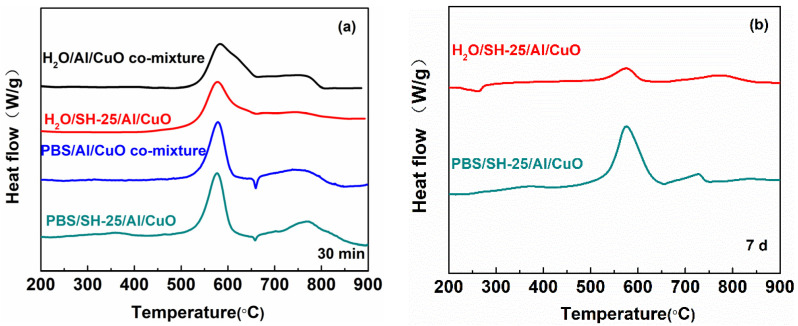
Heat release DSC plots of nanothermites. (**a**) Al and CuO assembled for 30 min. (**b**) Al and CuO assembled for 7 d.

**Table 1 ijms-23-08054-t001:** The total heat energy (ΔH) released by nanothermites.

Sample	30 min J/g	7 d J/g
H_2_O/Al/CuO comixture	1203	-
H_2_O/SH-25/Al/CuO	1277	323
PBS/Al/CuO comixture	1413	-
PBS/SH-25/Al/CuO	1785	1675

## Data Availability

The data presented in this study are available on request from the corresponding author.

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
