# Peer review of "Peptide Assembly of Al/CuO Nanothermite for Enhanced Reactivity of Nanoaluminum Particles"

_ijms, 2022, doi:10.3390/ijms23148054_

Round 1
Reviewer 1 Report
Manuscript ijms-1808818
The manuscript by Jin et al reports on the effect of a peptide on the reactivity features of Al/CuO nanotermites. The presence of the peptide-based surfactant during the Al/CuO nanocomposites enhances the stability of the nanoparticles towards the oxidative degradation, which deplete their thermal activity.
It is my opinion that the manuscript, although of interest, should not be accepted in this form, lacking some substantial parts for making it of interest for a broader arena of readers. The draft seems a list of interesting experimental results with no sound discussion.
For example, no explanation has been provided on the chemical nature of the effect of the presence of the surfactant/peptide on the stability of the systems investigated, which constitute the core of the work. More points are listed below.
1) Figure 2; insert brief explanation of the peaks in the caption.
2) page 3, paragraph 2.2, last lines: Describe peptide SH-25 and its properties, such as the cmc. The physico-chemical feature of the surfactant employed should be clearly reported and correlated to the effects obtained. Moreover, the concentration of the surfactant is not indicated in the text.
3) Figure 4: the caption is meaningless, not reporting a brief description of the plot.
4) Pag 5, line 7 from bottom: change “Various pH strengths were observed” to “A variation of pH was observed …”
5) Page 6, line 11 from bottom: replace “the size and zeta potential” by “the features”. Please briefly explain the significance of the zeta-potential on the features of the nanoparticles.
6) Page 7: please explain the type of structural change of the surfactant and the correlation with the binding affinity.
7) Page 8, line 3 after Table 1: change “acquired” with “observed”.
Reviewer 2 Report
The question remains for the authors, whether the sampling and the number of samples is robust enough to support the results.

Round 2
Reviewer 1 Report
This is a revised version of the manuscript by Jin et al, that I think is publishable in IJMS after fulfilling the following point.
Please indicate the critical micellar concentration (cmc) of the surfactant used, in order to assess if the effect on the stability of the AlCu composite is due to the formation inside the micellar core or it is just a surfacial electrostatic effect.
Minor point; line 139 correct the word "observied"
